# Correlation of Bronchoscopy and CT in Characterizing Malignant Central Airway Obstruction

**DOI:** 10.3390/cancers16071258

**Published:** 2024-03-23

**Authors:** Sukumar Kalvapudi, Hafiz M. Zubair, Rajesh Kunadharaju, Sajeer Bhura, Hiwot Mengiste, Musa Saeed, Arjun Saradna, Harshwant Grover, Gal Shafirstein, Sai Yendamuri, Nathaniel M. Ivanick

**Affiliations:** 1Department of Thoracic Surgery, Roswell Park Comprehensive Cancer Center, Buffalo, NY 14203, USA; sukumar.kalvapudi@roswellpark.org (S.K.); sai.yendamuri@roswellpark.org (S.Y.); 2Department of Pulmonary and Critical Care, University at Buffalo, Buffalo, NY 14203, USA; zubairiqbalian74@gmail.com (H.M.Z.); drrajesh125@gmail.com (R.K.); bhurasajeer@gmail.com (S.B.); hiwotmen@buffalo.edu (H.M.); musasaee@buffalo.edu (M.S.); arjunsaradna@gmail.com (A.S.); harshwan@buffalo.edu (H.G.); 3Department of Cell Stress Biology, Roswell Park Comprehensive Cancer Center, Buffalo, NY 14203, USA; gal.shafirstein@roswellpark.org

**Keywords:** central airway obstruction, bronchoscopy, CT, lung cancer

## Abstract

**Simple Summary:**

Malignant Central Airway Obstruction (MCAO) is a serious complication of lung cancer, leading to increased morbidity and mortality. MCAO can present in three major ways, with tumor pressing in on the airways from outside, tumor growing inside the airway, or a combination of both. While doctors often use bronchoscopy to confirm and assess MCAO, CT scans are used as an initial screening tool for treatment planning purposes. We studied 108 patients to see how well CT scans matched up with bronchoscopy results. CT scans correctly identified MCAO in most cases but were often discrepant in their estimates of severity by as much as 25% as compared to bronchoscopic evaluation. We found that CT scans were also limited in their ability to predict the type of airway blockage encountered on bronchoscopy. We conclude that, while CT scans are a useful screening tool for MCAO, bronchoscopic confirmation is recommended.

**Abstract:**

Background: Malignant Central Airway Obstruction (MCAO) presents a significant challenge in lung cancer management, with notable morbidity and mortality implications. While bronchoscopy is the established diagnostic standard for confirming MCAO and assessing obstruction subtype (intrinsic, extrinsic, mixed) and severity, Computed Tomography (CT) serves as an initial screening tool. However, the extent of agreement between CT and bronchoscopy findings for MCAO remains unclear. Methods: To assess the correlation between bronchoscopy and CT, we conducted a retrospective review of 108 patients at Roswell Park Comprehensive Cancer Center, analyzing CT and bronchoscopy results to document MCAO presence, severity, and subtype. Results: CT correctly identified MCAO in 99% of cases (107/108). Agreement regarding obstruction subtype (80.8%, Cohen’s κ = 0.683, *p* < 0.001), and severity (65%, Quadratic κ = 0.657, *p* < 0.001) was moderate. CT tended to equally overestimate (7/19) and underestimate (7/19) the degree of obstruction. CT was also poor in identifying mucosal involvement in mixed MCAO. Conclusions: CT demonstrates reasonable agreement with bronchoscopy in detecting obstruction. Nevertheless, when CT indicates a positive finding for MCAO, it is advisable to conduct bronchoscopy. This is because CT lacks reliability in determining the severity of obstruction and identifying the mucosal component of mixed disease.

## 1. Introduction

Malignant Central Airway Obstruction (MCAO) describes blockage of the trachea, main stem bronchus, and/or bronchus intermedius by tumorous growths [1]. This obstruction can stem from any primary or metastatic intra-thoracic malignancy, with lung cancer being the most common cause [2].

In a systematic epidemiologic study of MCAO diagnosed in lung cancer patients at Roswell Park Comprehensive Cancer Center, we observed that 17% of patients presented with MCAO at the time of lung cancer diagnosis. Furthermore, 8.2% of patients developed incident MCAO within five years [3]. A similar investigation conducted in England reported prevalence of 13% and incidence of 5%, respectively [4]. Our analysis also revealed that MCAO served as a significant predictor of mortality, even when adjusting for gender, histology, stage, and treatment history (Hazard ratio = 1.702, *p* < 0.001). Notably, the impact of MCAO on mortality exceeded that of advancing from stage 1 to stage 2 disease (HR = 1.58, *p* = 0.03). Similarly, Daneshvar et al. identified that central airway obstruction (CAO) detected on the index CT scan independently correlated with an increased hazard of death (HR = 1.78, *p* = 0.001), after accounting for age, gender, and disease stage. Although the cause of death was not tracked in either study, data from two other articles suggest that in up to 40% of patients with lung cancer, the cause of death is directly related to burden of local disease, many by obstruction [5,6]. This matches with our clinical experience of the problem.

Due to the significant disease burden of MCAO and associated morbidity, identifying and accurately characterizing MCAO is essential. Diagnosing MCAO, however, is challenging due to its diverse manifestations, ranging from severe symptoms, such as resting dyspnea, stridor, and hemoptysis, to milder presentations, such as cough and exertional dyspnea when less obstruction is present [7]. Patients frequently present with non-specific respiratory symptoms, which may be mistaken for pulmonary manifestations of asthma, COPD, or malignant disease progression [8].

With easy access and a non-invasive nature, cross-sectional imaging modalities, such as CT scans or PET CT images, are commonly used for initial identification of MCAO [9]. Additionally, CT scan findings are commonly used to predict severity and subtype of obstruction. Subtypes of CAO are classified based on whether the tumor is confined to the airway lumen, termed “intrinsic” (endoluminal), or if the tumor obstructs the airway due to mass effect without an endoluminal component, referred to as “extrinsic” (extraluminal). Most cases fall into the “mixed” category, involving elements of both intrinsic and extrinsic involvement [1]. CAO severity can be categorized into mild obstruction (25–50%), moderate obstruction (51–75%), and severe obstruction (>75%) [3]. Details such as severity and subtype are crucial as they enable proceduralists to anticipate appropriate treatment strategies. Intrinsic disease can be managed with ablative modalities, while extrinsic-only disease is predominantly addressed with stenting, along with newer therapies, such as interstitial photodynamic therapy (PDT) or intra-tumoral chemotherapy [10,11,12,13]. Knowledge of disease severity facilitates planning for potential hypoxia or hemodynamic instability [14,15]. If CT is employed to screen for MCAO cases, it would be valuable to ascertain how accurately CT findings correlate with bronchoscopy findings in characterizing MCAO. Currently, the correlation between CT scan findings of MCAO and bronchoscopy findings remains uncertain. Given the severity and high risk of MCAO, it is of significant clinical importance to assess the role of CT in identifying and characterizing this disease.

To assess the relationship between CT and bronchoscopy, we performed a retrospective observational study of patients with primary lung cancer with MCAO who received treatment at Roswell Park. Our aim was to evaluate the accuracy of cross-sectional imaging in identifying and characterizing MCAO when compared to bronchoscopy. Specifically, we assessed how well cross-sectional imaging aligned with bronchoscopy findings in identifying the subtype of MCAO and quantifying the degree of airway obstruction.

## 2. Methods

### 2.1. Cohort Selection

We utilized two distinct patient cohorts to investigate the correlation between central airway obstruction, as evaluated by bronchoscopy and cross-sectional imaging. This retrospective observational study employed the following approach in selecting cohorts.

### 2.2. CT Identified Cohort

The initial cohort, denoted as the “CT Identified Cohort”, was established based on our previous study aimed at assessing the incidence and prevalence of MCAO. In this study, a retrospective analysis of cross-sectional images was conducted for all patients diagnosed with primary lung cancer at Roswell Park Comprehensive Cancer Center (RPCCC) in 2015 to identify cases of MCAO (*n* = 114). Only patients with accompanying bronchoscopy data (*n* = 54) within three months of diagnosis were included in the present analysis. Bronchoscopies during this time were performed by two interventional pulmonologists and six thoracic surgeons.

### 2.3. Bronchoscopy Identified Cohort

Due to the limited number of patients in the first cohort, we identified an additional set of patients to increase the sample size. Patients diagnosed with primary lung cancer and airway obstruction who underwent either diagnostic or therapeutic bronchoscopy, performed by a single interventional pulmonologist (Nathaniel Ivanick) at RPCCC between August 2020 and June 2023, were selected as the second cohort. We opted to utilize bronchoscopies solely performed by a single proceduralist to assess and potentially mitigate variability in the visual estimation of the degree and subtype of obstruction. This approach was chosen to address potential discrepancies that might arise if the assessments were performed by multiple proceduralists, as was the case in the CT-identified cohort. This cohort was termed the “Bronchoscopy Identified Cohort.” Only patients with corresponding cross-sectional imaging data within three months of bronchoscopy were included in the final analysis (*n* = 55).

### 2.4. Variables and Measurement

Cross-sectional images from both cohorts were utilized to ascertain the presence, severity, and subtype of MCAO. Both contrast and non-contrast CT scans were used. All scans were reviewed by pulmonary physicians, including a board-certified interventional pulmonologist, a board-certified pulmonologist and second- or third-year pulmonary fellows. All physicians received training in scan interpretation, including didactic lectures and at least one hour of supervised CT scan interpretation. Each scan was independently reviewed by both a member of pulmonary attending staff and a pulmonary fellow. CT scans with a slice thickness of 5 mm were utilized, and images were examined in axial, sagittal, and coronal views. For the sagittal and coronal reformats, a standard soft tissue kernel was applied, acquiring images axially and subsequently performing standard planar reconstructions. To enhance image quality, we utilized a post-processing option on PACS specifically tailored for evaluating lung structures. The degree of obstruction on CT was estimated by comparing the greatest degree of luminal narrowing to the greatest proximal airway opening within the same airway segment [16,17]. In cases of diffuse narrowing, older CT scans were referenced to estimate the severity of obstruction. Severity of obstruction was then rated as mild (25–50%), moderate (51–75%), and severe (>75%). The subtype of obstruction on CT was characterized as extrinsic (compression from outside with no luminal margin thickening), intrinsic (presence of tumor in airway lumen without luminal margin thickening), or mixed (irregular border along luminal margin).

Bronchoscopy reports were independently reviewed by a board-certified interventional pulmonologist and two second- or third-year pulmonary fellows. All bronchoscopies included in this series were conducted via rigid bronchoscopy, utilizing either the Bryan^®^ rigid bronchoscope or tracheoscope. Orange (outer diameter 13.2 mm/inner diameter 12.2 mm) or Black (outer diameter 12 mm/inner diameter 11 mm) bronchoscopes were utilized, with ventilation facilitated by a Monsoon Jet Ventilator. Unfortunately, consistent records of the rigid bronchoscope size and jet ventilator settings were not available. For flexible bronchoscopy, the Olympus^®^ BF-1TH190 (Olympus America, Center Valley, PA, USA) or a prior equivalent was utilized. All reviewers received at least one hour of individual training, including supervised interpretation on airway obstruction subtype. In instances where the subtype could not be agreed upon, Nathaniel Ivanick assigned the final subtype of airway obstruction. Details on presence of obstruction, degree of obstruction and type of obstruction were collected. Given the retrospective nature of our study, we relied on visual measurements provided by endoscopists or, in the absence of such measurements, on visual estimates by reviewers for assessing the degree of airway obstruction. While no standardized criteria for endoscopic measurement were universally applied, a grading system indicating the severity as a percentage of luminal obstruction was generally available. Luminal obstruction was categorized by researchers into quartiles: 0–25%, 25–50%, 51–75%, and 76–100%. Subtype of airway obstruction on bronchoscopy was recorded by assessing the available bronchoscopy images. Subtypes of airway obstruction on bronchoscopy were recorded based on available bronchoscopy images, categorized as intrinsic (tumor within airway lumen distinct from luminal border), extrinsic (cartilaginous narrowing without tumor within lumen), or mixed (both visible tumor in airway lumen and cartilaginous narrowing). It is acknowledged that considerable variability exists in visual estimations among different proceduralists. Consequently, to mitigate this variability, the bronchoscopy identified cohort was specifically selected to include only bronchoscopies performed by a single proceduralist.

Severity and subtype of MCAO were compared between CT and bronchoscopy. Age, gender, race, stage of cancer, date of bronchoscopy and date of cross-sectional imaging data were collected for all patients.

### 2.5. Statistical Analysis

Statistical analyses were performed using IBM SPSS statistical software version 28.0.1.0. Patient demographic and disease data were presented as counts (percentages) for categorical variables and median (Interquartile Range) for continuous variables. To compare group differences between cohort 1 and cohort 2, we utilized independent t-tests for continuous variables and chi-square tests for categorical variables. Overall comparisons between CT and bronchoscopy were assessed using percentage agreement for the presence, type, and severity of obstruction. Cohen’s kappa (κ) was utilized to determine interrater agreement regarding the type of obstruction, while quadratic weighted kappa (κ) was employed to compare agreement between CT and bronchoscopy concerning the degree of obstruction. A significance level of *p* < 0.05 was considered statistically significant.

## 3. Results

In total, 108 patients were identified to have MCAO on cross-sectional imaging. Demographic distribution was similar across the two cohorts, as detailed in Table 1 Overall, patients were a median age of 65, with a higher likelihood of being female (58/108, 54%), white (92/108, 85%), and having Stage IV disease (63/108, 58%).

On average, patients underwent bronchoscopy at a median of 13 days after cross-sectional imaging. Bronchoscopy identified airway obstruction in 107 out of 108 patients (99% agreement). One patient with mild extrinsic airway obstruction detected on CT did not show obstruction on bronchoscopy. Patients in the bronchoscopy-identified cohort were more likely to have severe obstruction when compared to those in the CT-identified cohort (67% vs. 46%). In terms of disease subtype as assessed by bronchoscopy, the CT-identified cohort showed a higher prevalence of mixed disease (17/50, 34%), whereas the bronchoscopy-identified cohort exhibited a higher prevalence of intrinsic disease (37/54, 69%). However, these differences did not reach statistical significance (*p* = 0.065).

104 patients had details regarding subtype of obstruction (Table 2). There was moderate agreement regarding subtype of obstruction (84/104, 80.8%, Cohen’s κ = 0.683, *p* < 0.001) between bronchoscopy and CT [18]. Excellent agreement was noted between the two modalities when bronchoscopy identified obstruction as extrinsic (24/27, 89%) or intrinsic (15/17, 88%). The most significant area of discrepancy arose when obstruction was identified as mixed during bronchoscopy. CT tended to classify most of these cases as mixed (45/60, 75%), but a notable proportion were categorized as extrinsic (13/60, 22%).

Data on the severity of obstruction were available for 103 patients (Table 3). Moderate agreement was observed between bronchoscopy and CT regarding the degree of obstruction (67/103, 65%, Quadratic κ = 0.657, *p* < 0.001). When discrepancies occurred between the two methods, they typically differed by one degree. For instance, among patients with mild obstruction on bronchoscopy, 10/23 were classified as mild on CT, 11/23 as moderate, whereas only 2/23 were interpreted as severe obstruction on CT. Similarly, among patients with severe obstruction on bronchoscopy, 52/61 CT reads agreed, 8/61 noted moderate obstruction on CT, and only 1/61 was categorized as poor obstruction. In cases of moderate obstruction on bronchoscopy, CT tended to equally overestimate (7/19) and underestimate (7/19) the degree of obstruction.

We conducted separate analyses of the bronchoscopy-identified cohort and the CT-identified cohort to assess whether the variability in visual assessment among different proceduralists affected the outcomes. Our findings indicate that both cohorts yielded very similar results in terms of CT vs. bronchoscopy agreement, despite the variance in bronchoscopic assessment. Regarding the subtype of obstruction, the bronchoscopy-identified cohort had a percentage of agreement of 81% (44/54, Cohen’s κ = 0.667, *p* < 0.001) between the two modalities, and the CT-identified cohort had a percentage of agreement of 80% (40/50, Cohen’s κ = 0.685, *p* < 0.001) (Appendix A). For the degree of obstruction, the bronchoscopy-identified cohort exhibited a percentage of agreement of 69% (37/54, quadratic κ = 0.692, *p* < 0.001), while the CT-identified cohort demonstrated a percentage of agreement of 61% (30/49, quadratic κ value = 0.607, *p* < 0.001) (Appendix A).

## 4. Discussion

In this study, we systematically compared CT imaging with bronchoscopy for assessing MCAO for the first time. Our findings confirm the notion that CT is accurate in detecting presence of MCAO. However, while CT is commonly utilized to characterize MCAO subtype and severity, our results reveal a few key differences between CT and bronchoscopy. Notably, CT often disagreed with bronchoscopy on the severity airway obstruction and mischaracterized mixed subtype of airway obstruction. Moving forward, it is important to consider the clinical implications of these differences.

CT has firmly established itself as the gold standard for screening airway obstruction due to its widespread availability and non-invasive nature [7]. Screening for MCAO is crucial as it sometimes presents with nonspecific symptoms, as discussed previously [7] Our findings reveal a strong correlation between CT and bronchoscopy in identifying the presence of MCAO. Out of 108 patients with MCAO identified on CT, 107 had confirmed airway obstruction on bronchoscopy (99%). This suggests that cross-sectional imaging is accurate in detecting MCAO. An important caveat to this statement is that readers in our study were made aware of the possibility of MCAO and were looking at it. A previous study by Harris et al. reported that radiologists miss 30% of MCAO diagnoses when radiologists are not made aware of the possibility [19]. Another report suggested that a dedicated review of central airways increased the proportion of CAO detection by 30% [4]. The contrapositive of this statement is that, without a dedicated review, MCAO might be missed up to 30% of the time, in agreement with Harris et al. Future studies should be performed in a prospective manner to determine sensitivity and specificity of CT in detecting MCAO. One approach to address the challenge of radiology non-recognition is by incorporating a checklist of areas that should be commented on when examining a chest CT, including airway obstruction. Artificial intelligence (AI) tools could play a pivotal role in flagging suspicious CT scans for further review, providing an additional layer of support in identifying cases of MCAO. Our group is currently exploring the use of AI to increase identification of MCAO.

There was moderate agreement between bronchoscopy and CT in quantifying the degree of obstruction (65% agreement, Quadratic κ = 0.657, *p* < 0.001). Typically, when there was disagreement, it was between mild and moderate obstruction or moderate and severe obstruction. Few patients were categorized as having severe obstruction on CT (2/23, 9%) when mild obstruction was present on bronchoscopy, or mild obstruction on CT (1/61, 2%) when severe obstruction was present on bronchoscopy (Figure 1). In cases of moderate obstruction, CT tended to equally overestimate (7/19) and underestimate (7/19) the degree of obstruction. Several factors contribute to the discrepancy in the degree of obstruction between the two modalities. Bronchoscopists typically estimate the degree of obstruction visually, a subjective and variable method lacking standardization. The use of a fisheye lens in bronchoscopy may lead to an overestimation of obstruction severity, especially with variations in distance [20]. Jet ventilation, which may be used during rigid bronchoscopy lacks the ability to create significant positive end-expiratory pressure which could hold airways open more than positive pressure ventilation. This could result in a smaller airway caliber due to bulging of the posterior membrane. Additionally, the timing of the respiratory cycle during which the CT image is captured plays a role. Non-contrast CT scans are obtained during maximal inhalation, which may not accurately reflect conditions observed during therapeutic bronchoscopies. Contrast-enhanced CT scans obtained during maximal exhalation could better approximate the conditions encountered during bronchoscopy, although a direct comparison of the accuracy of the two methods has not been studied. In summary, imaging modalities introduce variability based on specific scan characteristics, while bronchoscopy introduces variability based on distance and visual estimation.

Clinical judgment regarding treatment decisions is typically informed by a combination of clinical symptoms, CT imaging, and bronchoscopy findings. It is essential to recognize that each piece of information contributes to the overall assessment, and none should be considered in isolation. Nevertheless, bronchoscopy findings often carry more weight in treatment decisions, as most bronchoscopists tend to prioritize them over CT findings when determining the need for therapy. Considering these factors, our study suggests that CT mischaracterizes degree of airway obstruction by as much as 25%. While overestimation may not be detrimental, underestimation is concerning, as cases of more severe obstruction may be overlooked. Therefore, we recommend performing bronchoscopy whenever CT reveals airway obstruction, irrespective of the estimated degree, as it may potentially uncover more severe disease.

As mentioned previously, treatment planning for MCAO depends on its subtype, with ablation for intrinsic disease, stenting for extrinsic disease, and a combination of both for mixed disease [21]. Therefore, accurate identification of the type of airway obstruction is crucial in treatment planning. Treating extrinsic disease remains a clinical challenge. These tumors are typically not resectable. High dose curative radiotherapy has been associated with significant adverse events, including bleeding, fistulas and fibrotic airway obstruction, while a lower dose of ionizing radiation only improves atelectasis in a minority of cases [22,23,24,25]. The proximity of these tumors to pulmonary artery, and often invasion in the trachea and main stem bronchus, prevent effective use of thermal ablation. We have reported the potential benefit of ablating these tumors with computer-based image-guided interstitial PDT (I-PDT) using endobronchial ultrasound (EBUS) with transbronchial needle for fiber insertion [12]. Interstitial photodynamic therapy (PDT) is a technique for applying photodynamic therapy (PDT) to internal tumors using light delivered via fibers inserted percutaneously. In this approach the EBUS is utilized for diagnosis and image-guided therapy. This therapy can be added to standard of care stent placement and systemic therapies. The EBUS guided I-PDT also has the potential to positively impact anti-tumor immunity. A follow-up Phase I/II trial is ongoing to further test the potential benefit of this new therapy for patients with extrinsic and mixed obstruction.

Our analysis revealed moderate to good agreement between bronchoscopy and CT regarding the subtype of obstruction (81% agreement, Cohen’s κ = 0.683, *p* < 0.001). There was excellent concordance between the two modalities when subtype was identified as extrinsic (15/17, 88%) or intrinsic (24/27, 89%) on bronchoscopy. The primary area of disagreement was that a number of cases were characterized as extrinsic disease on CT (13/60, 22%) when they were mixed on bronchoscopy. One possible explanation is that bronchoscopy may unveil features not easily discernible on CT scans. It is possible that extrinsic and mixed obstruction exist on a spectrum, with luminal wall edema and luminal wall tumor infiltration existing in obstruction that is more heavily extrinsic, but still with some mixed component [26]. Bronchoscopy’s ability to detect these nuances enables a classification of mixed subtype, which may not be as evident on standard 5 mm CT scans (Figure 2). Higher resolution CT scans with virtual bronchoscopy reconstruction might enhance the identification of such subtleties [27]. In a previous study by Naidich et al., CT scans were found to accurately identify focal airway lesions with 90% accuracy. However, the study concluded that axial CT scans were inaccurate in predicting whether the abnormality would be endobronchial, submucosal, or extrinsic [28]. Finkelstein et al. conducted a study involving 44 patients using super high-resolution CT scanning, revealing sensitivities for the detection of endoluminal, obstructive, and mucosal lesions at 90%, 100%, and 16%, respectively [29]. Another retrospective study by Mamatha et al., involving 426 patients, found the sensitivity of CT in detecting abnormal mucosa, endobronchial masses, and extrinsic compression to be 27.50%, 89.00%, and 59.52%, respectively [30]. Although these studies primarily focused on identifying focal airway abnormalities in both healthy and diseased patients, rather than specifically characterizing central airway obstruction, a consistent trend emerges indicating poor characterization of mucosal lesions. This aligns with our observations that mixed diseases with predominantly extrinsic component might be erroneously interpreted solely as extrinsic on CT due to its inability to detect mucosal abnormalities. Mixed disease should be less prone to being misinterpreted as intrinsic disease, since intrinsic airway compression should be recognizable on both imaging and bronchoscopy, consistent with our findings. Based on the above findings, we recommend exercising caution when extrinsic disease is identified on CT scans, as there is a risk of overlooking potential intrinsic disease components. We advocate for maintaining flexibility in treatment approaches, particularly by considering bronchoscopy for further evaluation when extrinsic disease is suspected based on CT findings.

## 5. Limitations

Our study has several limitations which may affect its generalizability of results to a broader population. First, it is a retrospective single-center study and may not fully capture the diversity of cases encountered in different settings. Second, only a portion of the patients from the CT identified cohort had bronchoscopies that we could include in this study. It is likely that this selected a more severe patient population than otherwise expected. We exclusively analyzed patients with primary lung cancer, and findings may differ in cases of metastatic cancers. Both contrast-enhanced and non-contrast scans were utilized in the study, and we did not analyze potential differences between the two modalities. We are currently planning a future study to address this issue. Most CT imaging utilized had a slice thickness of only 5 mm. As it was a retrospective study, we did not have the ability to reformat the CTs to allow for high resolution or thin slices. Higher resolution CT scans with virtual bronchoscopy reconstruction may offer a more accurate estimate of bronchoscopy findings, a hypothesis we plan to test in future studies. Readers of CT scans were explicitly instructed to focus on identifying airway obstruction, introducing a bias toward increased detection. Additionally, for documenting the degree and subtype of obstruction on bronchoscopy, we relied on visual estimates provided by endoscopists or on visual estimates by reviewers. No standardized criteria regarding assessment of such were used. We plan to address this limitation in a future prospective study, focusing on improving measurement accuracy and documenting ventilation details. Despite these limitations, our study offers valuable preliminary insights. Future research with larger, more diverse cohorts and a prospective design could further bolster the robustness of our findings.

## 6. Conclusions

To our knowledge, this study represents the first comparison of cross-sectional imaging to bronchoscopy in terms of MCAO characterization. While CT reliably identifies the presence of airway obstruction, this is contingent upon specific attention being paid to the airways. Improved methods for identifying MCAO on screening CTs, such as AI-based identification, are necessary to minimize cases of missed obstruction. We recommend performing bronchoscopy following all cases of MCAO diagnosed on CT, as CT may underestimate or overestimate degree of obstruction. CT imaging demonstrated limited capability in identifying mucosal involvement in mixed subtype of central airway obstruction (CAO), thus treatment planning solely based on CT findings should be avoided. Future prospective studies are warranted to evaluate the utility of high-resolution CT scans with 3D reconstruction in characterizing MCAO.

## Figures and Tables

**Figure 1 cancers-16-01258-f001:**
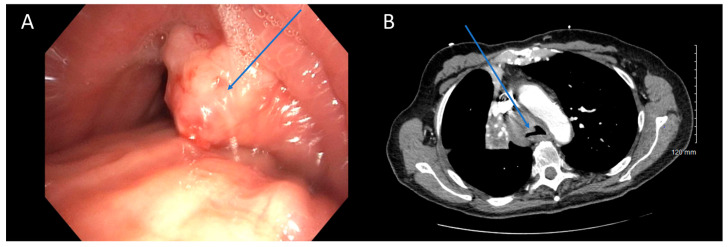
CT Underestimating Degree of Obstruction Compared to Bronchoscopy. Bronchoscopy (**A**) and CT (**B**) images depicting an endobronchial tumor in the distal trachea of a patient. The severity of obstruction appears to be greater when observed bronchoscopically compared to CT imaging. Arrows (in blue) highlight the tumors in the figure.

**Figure 2 cancers-16-01258-f002:**
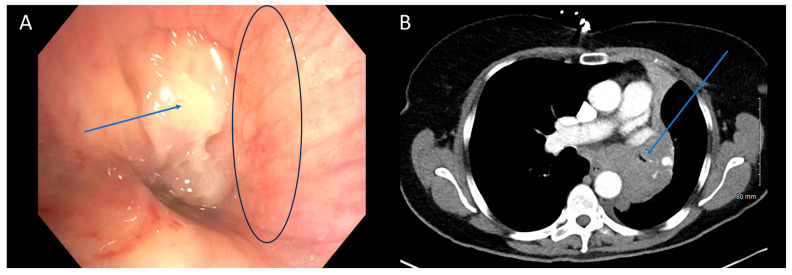
CT is Poor at Identifying Mucosal Component of Malignant Central Airway Obstruction. Bronchoscopic (**A**) and CT (**B**) images illustrating malignant central airway obstruction in the right main stem bronchus of the same patient. The CT image suggests primarily extrinsic obstruction, whereas bronchoscopy reveals a combination of extrinsic and intrinsic disease (mixed subtype). Arrows in blue indicate the tumors, while the black circle in (**A**) highlights the mucosal edema visible during bronchoscopy.

**Table 1 cancers-16-01258-t001:** Demographics of the CT Identified Cohort and the Bronchoscopy Identified Cohort.

Variable	Overall (*n* = 108)	CT Identified Cohort (*n* = 54)	Bronchoscopy Identified Cohort (*n* = 54)	*p*-Value
Median Age (Median, IQR)	65.5 (59–72.75)	65 (55.75–72.75)	66 (59–73)	0.514
Gender				0.7
Male	50 (46%)	26 (48%)	24 (44%)	
Female	58 (54%)	28 (52%)	30 (56%)	
Race				0.058
White	92 (85%)	50 (93%)	42 (78%)	
Black	8 (7%)	3 (5%)	5 (9%)	
Unknown	8 (7%)	1 (2%)	7 (13%)	
Stage				0.571
1	4 (4%)	1 (2%)	43(6%)	
2	9 (8%)	6 (11%)	3 (6%)	
3	28 (26%)	15 (28%)	13 (24%)	
4	63 (58%)	32 (59%)	31 (57%)	
Unknown	4 (4%)	0 (0%)	4 (7%)	
Degree of Obstruction (Measured by Bronchoscopy) *				0.271
Mild (25–50%)	23 (21%)	13 (27%)	10 (19%)	
Moderate (51–75%)	19 (18%)	11 (22%)	8 (15%)	
Severe (>75%)	61 (57%)	25 (51%)	36 (67%)	
Type of Obstruction (Characterized by Bronchoscopy) **				0.065
Extrinsic	17 (16%)	10 (20%)	7 (13%)	
Intrinsic	60 (56%)	23 (46%)	37 (69%)	
Mixed	27 (25%)	17 (34%)	10 (19%)	
CT to Bronchoscopy Interval (Median, IQR)	13 days (7–27)	14 days (8–28)	10.5 days (5–26)	0.233

Details of median age, sex, race, stage of cancer, degree of obstruction as measured by bronchoscopy, type of obstruction as characterized by bronchoscopy, and CT to Bronchoscopy interval in days. Group differences between cohort 1 and cohort 2 were calculated using independent *t*-tests for continuous variables and chi-square tests for categorical variables. The *p*-values for the group differences are given in the last column. CT = Computed Tomography, SD = Standard Deviation. * *n* = 103; ** *n* = 104.

**Table 2 cancers-16-01258-t002:** Subtype of Obstruction in CT vs. Bronchoscopy.

		Bronchoscopy			
		Intrinsic	Mixed	Extrinsic	Total (CT)
**CT**	Intrinsic	15	2	0	17
	Mixed	1	45	3	49
	Extrinsic	1	13	24	38
	Total (Bronchoscopy)	17	60	27	104

Details of subtype of obstruction as identified by CT vs. Bronchoscopy. Percentage of agreement between the two modalities was 80.8% (84/104). Cohen’s κ = 0.683, *p* < 0.001. CT = Computed Tomography.

**Table 3 cancers-16-01258-t003:** Degree of Obstruction in CT vs. Bronchoscopy.

		Bronchoscopy			
		25–50%	51–75%	76–100%	Total (CT)
**CT**	25–50%	10	7	1	18
	51–75%	11	5	8	24
	76–100%	2	7	52	61
	Total (Bronchoscopy)	23	19	61	103

Details of degree of obstruction as identified by CT vs. Bronchoscopy. Percentage of agreement between the two modalities was 65% (67/103). Quadratic κ = 0.657, *p* < 0.001. CT = Computed Tomography.

## Data Availability

Deidentified data are available on request. Data are not publicly available due to their containing information that could compromise the privacy of research participants.

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
