# Peer review of "Correlation of Bronchoscopy and CT in Characterizing Malignant Central Airway Obstruction"

_cancers, 2024, doi:10.3390/cancers16071258_

Round 1
Reviewer 1 Report
Comments and Suggestions for Authors
Dear Authors,
I read with interest the manuscript entitled "Correlation of Bronchoscopy and CT in Characterizing Malignant Central Airway Obstruction." The paper reports exciting data regarding the penetration of CT scans and bronchoscopies into the study of airway obstruction from central lung malignancies.
I ask for some clarifications:
1) Were all CT scans made with or without contrast dye? What image reconstruction algorithm has been used for axial and sagittal views?
2) The criteria to assess the endoscopic evaluation of obstruction is not defined. It is hard to understand if it depends mainly on the feeling/experience of the endoscopist (in this case, you could report a specific declaration from the endoscopist) or if an endoscopic calibre has ever been used. You could add more information regarding bronchoscopic approaches, rigid or flexible bronchoscopes used, and their diameter could be added in the text and in the explanatory data. Please revise this aspect. It is essential. Did you use a specific classification (like the Myer-Cotton in trachea obstructions) to address the bronchial stenosis?
3) line 13: is NI Nathaniel Ivanick?
4) Table 1: Even if there were no significant differences between groups' characteristics, please show the p-value when it is possible to compare.
5) Did you evaluate possible differences along the time frames? For example, splicing the bronchoscopy cohort in two (2020-2021 vs 2022-2023) and then considering whether there is a difference of agreement between CT and bronchoscopy in the two-time frame groups.
Reviewer 2 Report
Comments and Suggestions for Authors
The manuscript “Correlation of Bronchoscopy and CT in Characterizing Malignant Central Airway Obstruction” compares classification of Malignant Central Airway Obstruction (MCAO) using CT scans or bronchoscopy. A retrospective, single center observational study was conducted on patients with primary lung cancer and the aim was to evaluate the accuracy of identifying and characterizing MCAO in CT compared to bronchoscopy imaging. The authors concluded that “while CT scans is a useful screening tool for MCAO, bronchoscopic confirmation is recommend when CT indicates a positive finding for MCAO, it is advisable to conduct bronchoscopy”. The authors accurately point out the limitations of the study and carefully describe and discuss their findings using pertinent references. The statistical analysis is also well performed. There are only a few minor concerns:
1: Line 79: What is “interstitial PDT”? Please explain all acronyms.
2: Line 215: “An important caveat to this statement is that readers in our study were made aware of the possibility of MCAO and were actively looking at it”. Please change to “looking for it”.
3: Figure 1 and 2: Please indicate the tumor with arrows in A and B, for clarity. Not everyone is an expert in assessing images.
4: Reference 27: Please revise the format.
Comments on the Quality of English LanguageThe English is very good.
